# Deep Learning and Antibiotic Resistance

**DOI:** 10.3390/antibiotics11111674

**Published:** 2022-11-21

**Authors:** Stefan Lucian Popa, Cristina Pop, Miruna Oana Dita, Vlad Dumitru Brata, Roxana Bolchis, Zoltan Czako, Mohamed Mehdi Saadani, Abdulrahman Ismaiel, Dinu Iuliu Dumitrascu, Simona Grad, Liliana David, Gabriel Cismaru, Alexandru Marius Padureanu

**Affiliations:** 12nd Medical Department, “Iuliu Hatieganu” University of Medicine and Pharmacy, 400000 Cluj-Napoca, Romania; 2Department of Pharmacology, Physiology and Pathophysiology, Faculty of Pharmacy, “Iuliu Hațieganu” University of Medicine and Pharmacy, 400000 Cluj-Napoca, Romania; 3Faculty of Medicine, “Iuliu Hatieganu” University of Medicine and Pharmacy, 400000 Cluj-Napoca, Romania; 4Department of Computer Science, Technical University of Cluj-Napoca, 400114 Cluj-Napoca, Romania; 5Department of Anatomy, “Iuliu Hatieganu“ University of Medicine and Pharmacy, 400006 Cluj-Napoca, Romania; 6Fifth Department of Internal Medicine, Cardiology Rehabilitation, Faculty of Medicine, “Iuliu Hatieganu” University of Medicine and Pharmacy, 400347 Cluj-Napoca, Romania

**Keywords:** antibiotic resistance, antibiotic development, automated antibiotic discovery, adaptive resistance, computer-aided drug discovery, artificial intelligence (AI), machine learning, neural networks, deep learning, future of medicine

## Abstract

Antibiotic resistance (AR) is a naturally occurring phenomenon with the capacity to render useless all known antibiotics in the fight against bacterial infections. Although bacterial resistance appeared before any human life form, this process has accelerated in the past years. Important causes of AR in modern times could be the over-prescription of antibiotics, the presence of faulty infection-prevention strategies, pollution in overcrowded areas, or the use of antibiotics in agriculture and farming, together with a decreased interest from the pharmaceutical industry in researching and testing new antibiotics. The last cause is primarily due to the high costs of developing antibiotics. The aim of the present review is to highlight the techniques that are being developed for the identification of new antibiotics to assist this lengthy process, using artificial intelligence (AI). AI can shorten the preclinical phase by rapidly generating many substances based on algorithms created by machine learning (ML) through techniques such as neural networks (NN) or deep learning (DL). Recently, a text mining system that incorporates DL algorithms was used to help and speed up the data curation process. Moreover, new and old methods are being used to identify new antibiotics, such as the combination of quantitative structure-activity relationship (QSAR) methods with ML or Raman spectroscopy and MALDI-TOF MS combined with NN, offering faster and easier interpretation of results. Thus, AI techniques are important additional tools for researchers and clinicians in the race for new methods of overcoming bacterial resistance.

## 1. Introduction

Antibiotic resistance (AR) is a naturally occurring phenomenon consisting mostly of acquired adaptative mechanisms. These mechanisms aid bacteria to overcome and survive the aggression caused by antibiotics [1]. AR represents an important problem in present times due to several factors. A wide majority of the antibiotics currently in use have been discovered before the 1980s [2]. Some have been improved by chemical processes such as conjugation with other substituents (e.g., fluoroquinolones, which are a result of the addition of a fluoride atom to already existing quinolones). Although the pharmacokinetic and pharmacodynamic properties of the derivates are improved, these developmental processes do not solve the issue of AR because the main structure of the molecule remains the same.

Due to the current COVID-19 pandemic, funds have been redirected towards research, prophylaxis, and equipment to improve survival and to reduce the probability of complications amongst infected patients [3]. Moreover, with the attention shifting towards the current pandemic, it is estimated that antimicrobial resistance (AMR) has only accelerated during this period due to the extensive use of disinfecting agents, extensive prescription of antibiotics, and interrupting treatments of various other chronic conditions due to decreased access to the hospital [3].

Most antibiotics used today target the same few bacterial structures. Therefore, research should focus on developing molecules with innovative action mechanisms, which target new bacterial sites [4,5]. Antibiotic development faces economical and technical difficulties, with increased costs of research and marketing new antibiotics being some of the economic hardships [6,7,8]. Technical difficulties include lack of equipment, infrastructure, and trained personnel in some regions [6,7,8]. To overcome AR, international programs should be created and enforced globally. Such programs are currently developed with the aid of DL-constructed computer systems [9].

DL is a subtype of ML that utilizes artificial neural networks (ANN). Every ANN consists of multitudinous interconnected neurons, grouped into layers, having similar functionality to biological neurons. They receive an input, process the data, and generate an output signal. In the beginning, the program’s generated predictions will be highly inaccurate. The parameters linking the NN, namely the weights and biases, are thus far unrefined to form an appropriate processing pathway. However, the neural network architecture constantly evolves to form the correct pattern recognition [10].

The core principle is that the weights and biases’ values are self-evaluated with every input. Therefore, the more qualitative and heterogeneous data the network receives, the more precise its predictive power becomes. This self-evaluation arises via comparing the desired output and the output the algorithm produces. Based on this feedback, the parameters acquire increasingly authentic values, new patterns assemble, and the final form of the algorithm is constructed [10].

While ML still utilizes human intervention, when the desired output is different from the generated output, DL can adjust itself to meet the required criteria, making it the preferred method [11]. Thus, it has seen a growth in popularity in the precision medicine field. The creation of personalized therapy protocols is achieved through predicting, identifying, and validating bioactive agents fitting a patient’s molecular background [10].

With the aid of DL, such molecules’ effectiveness and design, i.e., physicochemical properties and synthesis, are generated in a financially and time-effective manner. Furthermore, it eliminates toxic side effects to the personnel handling the substances [12].

For this reason, the aim of this narrative review is to present the role of DL applications in the field of AR.

## 2. History

Bacteria possess a collection of genes that grant them resistance to external aggressions. The resistance genes’ inheritance model and transcription levels define the bacterial resistome [1]. Bacteria often develop resistance to antibiotics, but even before antibiotics have been used, they developed resistance to other aggressors.

The primary cause of bacteria becoming resistant is the effect of environmental factors such as exposure to sunlight, soil components, water, and even anthropic ecosystems. All these factors form the environmental resistome which has appeared long before bacteria were exposed to antibiotics [13]. However, AR appeared before any human life form. Gene-sequencing studies have shown that there is no significant difference between the AR mechanisms of bacteria that exist today and ancient bacteria [14]. In addition, genes coding enzymes that grant AR have been proven to be present in bacteria found in the permafrost. These enzymes show a structural likeness to those found in modern bacteria [15,16].

## 3. Present

AR is a pressing issue in current times, with a concerning lack of awareness from the health practitioners and the general public. This frequently determines an unjustified self-prescribed usage of antibiotics, as well as an over-prescription of antibiotics by medical practitioners, with studies showing that half of the prescribed antibiotics are unnecessary [17].

Infection prevention control strategies are deficient in a number of countries, especially in developing ones. The main factors to be considered are the governing political parties, infrastructure, lack of knowledge on proper prophylactic measures, urbanization, and the general low level of education in relation to antibiotic intake [18].

Modern technologies could come to aid in impeding AR. As infrastructure develops, so do the quality and stability of the newer antibiotic formulas. Moreover, bioinformatics has become crucial in the development of in silico methods that aid researchers in studying antibiotic interactions.

### 3.1. Antimicrobial Peptides Testing

Ruiz Puentes et al., proposed a new DL model, called AMPs-Net, that successfully classified a set of 23,967 peptides from known databases as antimicrobial peptides (AMPs) or non-AMPs, assigning each AMP to its class: antibacterial, antiviral, antifungal, or antiparasitic. To test this DL model and discover new AMPs, they used a restriction enzyme to cut an *Escherichia coli* (*E. coli*) genome and generate a new peptide library. With the help of a neural network generated algorithm, they were able to identify 252 peptides out of 423,697 sequences created by the restriction enzyme by filtering these 252 peptides with antimicrobial activity they found four AMPs which were further analyzed. Molecular dynamics assays were used to describe the capability of these AMPs to penetrate membranes. In vitro evaluation of these AMPs concluded that three out of four peptides presented antimicrobial activity. RD-10, a peptide consisting of ten amino acids, showed promising bacteriostatic activity and it was considered a potential candidate for further preclinical examinations [19].

Li et al., used a DL-derived tool that predicted new AMPs derived from the *Rana catesbeiana* (bullfrog) genome. They tested the new acquired peptides against the World Health Organization’s (WHO) priority list. They concluded that 4 of the 16 new discovered AMPs were highly active against multiple species of bacteria from the organization’s priority list, including multi-drug-resistant (MDR) carbapenemase-producing (CPO) *E. coli*. The algorithm works by partitioning its training set, and the authors concluded that this novel approach is superior to other deep-learning models. It was speculated that the AMPs, with no effect on the four bacterial isolates used in this study, may be active on other microbes such as fungi or viruses [20].

Lin et al., created a DL-derived AMP predictor that used six physicochemical properties (PC6) to encode peptides based on their chemical structure. Furthermore, the team implemented this algorithm in a web application, AI4AMP, by training the DL model with more than 13,000 peptide sequences. This application was able to calculate a score, the AI4AMP score, which is corelated with the antibacterial activity; higher score represent possible candidates for more preclinical testing. The authors concluded that an ideal encoding DL-based model should consider protein features, such as the physicochemical properties of peptides. Good analysis and choice of parameters lead to a better performance of the prediction software, thus reducing the loss of information when encoding peptides for further processing [21].

### 3.2. Detection of AR Genes

Yu Li et al., developed a multi-task method based on DL to ease detection of AR genes (ARGs). The technology provides details about the antibiotic class for which resistance is granted, the resistance mechanism, and gene mobility (if the gene is innate or plasmid-acquired). For the beta-lactamase antibiotic family, the algorithm can tell which subclass the ARG accounts for. The model presented in this study was trained using a dataset that was compiled manually out of 7 pre-existing ones for the technology to capture the most relevant features associated with ARGs. The DL technology proposed in this study performed better than other known models in characterizing ARGs presents in the human intestinal microbiota [22].

The study conducted by Arango-Argoty et al., proposes two DL models which focus on the similarity distribution of sequences in the ARGs database. This technology addresses the limitations of the traditional “best-hit” methodologies which can have a significantly higher rate of false negative results (they classify many ARGs as not being ARGs.) The DL models were trained separately using a collection of three distinct databases (CARD, ARDB, and UNIPROT). The first DL model was focused on predicting ARGs directly by reading short DNA sequences, while the other DL model could read complete ARGs sequences and was therefore used to discover novel ARGs. This technology’s limitations are that it cannot predict AR granted by single nucleotide polymorphisms (SNPs) and it only recognizes ARGs if they are part of one of the groups used in the training of the algorithm [23].

Steiner et al., studied the evolution of antiretroviral therapies (ART) resistance by using the human immunodeficiency viruses (HIV) genome. The study uses the genome of HIV-1 and drug resistance assay results for 18 known ARTs to test three different ANN architectures. The genotype and phenotype data were obtained from Stanford University’s HIV Drug Resistance database. All three DL methods were trained and tested in all 18 data sets for ART resistance. The study compares the performance of three ML algorithms in testing drug resistance and shows a higher classification performance of the NN algorithms [24].

### 3.3. Other Measures to Decrease AR

Nevertheless, prophylaxis is desired instead of treatment. Therefore, knowledge on proper hygiene is mandatory for preventing bacterial infections. Appropriate hand hygiene alongside food safety measures can reduce the number of infections and lead, in time, to a lower antibiotic usage [25]. Hospital-acquired multi-resistant bacteria pose a threat to patients and require extensive antibiotic treatments. Thus, proper disinfection and sterilization procedures represent helpful measures against nosocomial infections [26]. In addition, resistant bacteria hotspots should be located and eliminated to prevent further development of bacterial reservoirs.

The rise in urbanization levels leads to increased pollution and overcrowded areas and, together with other factors such as the individual’s susceptibility and pathogen’s virulence, these facilitate chronic respiratory tract infections. Recurrent infectious episodes lead, in time, to the evolution of multi-resistant bacteria [27].

Usage of antibiotics in the fodder fed to livestock affects humans as well. Animal excrements containing traces of antibiotics determine an environmental selection pressure. The consequence is the appearance of drug-resistant bacteria, which cause serious human infections. In addition, animals can be responsible for AR through mobile genetic elements (MGE), such as transposons. For example, the mcr-1 transposon was responsible for the development of AR in swine farms in Northeastern China. When mcr-1-positive *E. coli* was frequently isolated from meat in China, its prevalence in animal food attracted negative public attention [28].

However, the antibiotic-producing industry is more focused on generic drugs that resemble established, original molecules, instead of trying to develop entirely new formulas. Instead of creating innovative antibiotics with unique action mechanisms, the trend is to produce virtually identical chemical drugs [13].

Tourism is another possible factor for the emergence of AR bacteria. With expanding tourism, there is also a higher chance of people becoming vectors for bacteria. Through this extensive spread, bacteria are responsible for creating new reservoirs that can transform into endemic outbreaks [29].

## 4. Future Perspectives

Due to the increasing number of multi-resistant bacteria, new approaches regarding antibiotics development should be considered. Based on some important progresses made in the recent years, beginning with the introduction of AI in antibiotics development, some new areas of antibiotic discovery have started to yield encouraging results (Figure 1).

### 4.1. Critical Findings Concerning AI in Antibiotics Development

Initially, AI was designed to carry out simple tasks and was generally used in the industry to create and assemble parts, as well as to reduce costs. Continuous developments in this field are currently shaping new opportunities for AI to be used in the field of AR for the design of new therapeutic agents in order to tackle a potential future crisis [11].

It can take up to 10–12 years for drugs to reach the market, including antibiotics. The first period of the drug development process usually ranges from 2–4 years. In this period, new active substances are investigated from a chemical point of view. Only a few manage to pass this step before being tested in animal studies to discover any toxicological effects [30]. AI can shorten the preclinical phase by rapidly generating many substances based on algorithms created by ML techniques. These algorithms predict the antibiotic efficacy for each generated molecule by analyzing large sets of data, making the whole early drug discovery process much faster [31].

ML is a technology that analyzes models and builds up algorithms depending on the data which is initially introduced in the database. These computational methods can adapt and become better based on previous experience. The predictive power of an algorithm grows proportionally to the quantity and quality of the initial dataset and allows us to explore a vast amount of information beyond the reach of traditional approaches.

NNs are also a subset of ML which are rapidly emerging. NNs are inspired by the human brain, in which neurons obtain input information, process it, and transform it to a specific output signal. Likewise, NN can recognize patterns and can learn from examples before processing incoming data and generating predictions. Weights are assigned for each input variable, and by constantly adjusting in the processing pathway, NN are able to improve their accuracy over time, which makes them a powerful asset to AI [32]. DL algorithms are basically NN which have more than three hidden layers.

### 4.2. Drug Repurposing Testing

An important aspect in which AI technologies are valuable is drug repurposing testing. By combining the above-mentioned methods, already-known approved or unapproved substances can be screened for antimicrobial effects. With the help of computer-generated simulations, interactions between these substances and bacterial structures can be studied. A good example of drug repurposing is aspirin, currently being used as an antiaggregant in addition to its anti-inflammatory effects [33]. The beta-blocker propranolol is also used nowadays in the treatment of infantile hemangioma, proving that drug repurposing can also be an efficient method in antibiotic development programs [34]. Given that many new substances never reach clinical testing, drug repurposing can be very helpful and AI techniques can be beneficial by helping with scanning and filtering the large number of known drugs as well as drugs that undergo preclinical and clinical testing for other indications.

### 4.3. Discovery of Antibiotic Peptides

ML is also used in the discovery of antibiotic peptides [35]. Peptides with antimicrobial activity are also widely distributed in different lifeforms where they play an essential role as part of the innate immune system. The most known mammalian AMPs are cathelicidins and defensins, acting as human host defense peptides (HDPs). They are secreted in different parts of the organism such as the skin, eyes, respiratory tract, lung, and intestine. Their main role is to act fast by being part of the innate immune system and provide a broad-spectrum protection against invading pathogens [36]. Microorganisms like bacteria and fungi also produce AMPs which help them fight against each other. AMPs are simple peptides without a complex 3D structure like large proteins, however their cost of production is expensive, at about USD 100–1000 for 1 mg of AMP, and they are only produced in lab conditions for experiments [37].

The most widely known mechanism through which AMPs kill microbes is osmotic shock, which occurs either through the formation of pores, or paving as carpet on the membrane surface to weaken membrane integrity [35,38].

Unlike most conventional antibiotics, which have specific functional or structural targets, AMPs act directly on the microorganisms, often causing cell lysis, or modulate the host immunity to enhance defense against microorganisms [39]. Moreover, they act faster than conventional antibiotics [19], have a narrower active concentration window for killing [40], and do not typically damage the DNA of their targets [20,41], though there have been studies describing how AMPs inhibit critical intracellular functions by binding to DNA, RNA, or intracellular proteins [42]. As a result, they do not induce resistance to the extent that is observed with conventional antibiotics [21]. Nevertheless, if bacteria are exposed to AMPs for extended periods of time, they can and do develop resistance even to peptide-based drugs, including the last-resort and life-saving drug, colistin [20,21].

The quantitative structure-activity relationship (QSAR) method, combined with ML techniques, was successfully used in 2009 by a group with the objective to discover new AMPs with antibacterial properties [43]. Their algorithm managed to predict and rank antimicrobial activity of 100,000 virtual peptides, with 94% of the 50 highest ranking peptides later being proved as being highly active against strains of multidrug-resistant *Pseudomonas aeruginosa* (*P. aeruginosa*), methicillin-resistant *Staphylococcus aureus* (MRSA), extended-spectrum β-lactamase-producing *E. coli* and *Klebsiella pneumoniae* (*K. pneumoniae*), as well as vancomycin-resistant *Enterococcus faecalis* and *Enterococcus faecium* (VRE) [43]. QSAR is a computational modeling method that finds relationships between the structural properties and biological effects of compounds. This method helps in antibiotic discovery by prioritizing the relevant active molecules to be further tested in animal studies, saving time and money [39]. The QSAR method shows a high potential when it comes to antibiotic discovery, especially when it is combined with ensemble methods which can improve predictability [39].

Generally, a molecule able to inhibit many different strains of the same bacterium will have a lower propensity than a molecule that inhibits a few strains. By following this thread, new QSAR approaches, such as the multi-tasking model for quantitative structure-biological effect relationships (mtk-QSAR), also called multi-target (mt-QSAR), are used to integrate different kinds of chemical and biological data, allowing the assessment of multiple biological activities against diverse biological systems. By screening large amounts of data, the descriptors can show which pattern is often associated with the desired conditions (for instance high antimicrobial activity against multiple Gram-negative bacteria and low cytotoxicity to human cells). In the end, known peptides are prioritized based on their similarity with these descriptors and other peptides can also be generated with the established set of rules. Only the combination of certain amino acids and the topological distances between them are essential for improving the antibacterial activity [44]. In the study conducted by Kleandrova et al., the study group generated a library formed by 10 peptides, all which exhibited high antibacterial activity against Gram-negative bacteria based on the molecular descriptors which they used to screen a data set containing 3592 peptides [44].

To this extent, the use of in silico models based on perturbation theory concepts and machine learning technologies (PTML) could measure the probability of a drug being active under certain conditions (protein, cell line, organism) [45]. The combination between PTML and mtc-QSAR models showed promising results in discovering multi-strain inhibitors. In another study, Kleandrova et al., showed that this approach proved to be helpful in identifying molecules that could extend antituberculosis (anti-TB) effect. Twelve molecular descriptors were chosen and their tendencies of variation were calculated, meaning how much these descriptors should vary in order for a molecule to enhance its anti-TB activity and versatility (ability to inhibit more than one *Mycobacterium tuberculosis* strain). For example, one descriptor analyzes the augmentation of the molecule brought by the hydrophobicity of any two atoms which are located three bonds from one another. The mtc-QSAR-EL model identified proven antituberculosis drugs after screening a dataset made up of 8898 agency-regulated chemicals (and investigational FDA-approved drugs). It also recommended compounds with high potential to be experimentally repurposed as antituberculosis (multi-strain inhibitors) agents [45].

NN-based algorithms are also used to identify and describe the bioactivity of each peptide sequence. For using recurrent neural networks (RNNs), representation methods together with architecture for each representation must be used to analyze the sequence of amino acids. Vectors are commonly used to describe essential properties of AMPs [19]. They are used to represent the order of amino acids in AMPs, for example by representing peptides as sequences of vectors, describing the presence of one of the 20 essential amino acids in the sequence. Alternatively, 1D vectors have been considered, where each amino acid in the sequence is codified by a number from 0 to 19. More recently, peptides have been reproduced with the aid of the word2vect denomination. However, sequential representation does not consider the interactions between amino acids and the atoms within each amino acid. Essential properties of AMPs must be included, such as amino acid composition or composition–transition–distribution, to make the vectors more descriptive [19].

### 4.4. Other Applications Combined with AI for Antibiotic Discovery

Lu et al., used the Raman spectroscopy combined with ML in order to depict AR status of *K. pneumoniae*. Raman spectroscopy is a powerful tool that requires a sample to generate a fingerprint with the chemical constituents of the sample. After training the neural network with the initial dataset consisting of 71 *K. pneumoniae* isolates, the algorithm was programmed to output a probability distribution across seven ARGs, two virulence genes, and the drug-resistant phenotypes (sensitive or non-sensitive), among 15 commonly used antimicrobial agents. This convolutional neural network (CNN) has proved to be better at predicting ARGs, virulence factors, and drug-resistant phenotypes compared to other past ML algorithms. It is also much faster than conventional antimicrobial-susceptibility testing (AST) methods which imply growing of bacterial colonies on agar plates in the presence of antibiotics and the study group. When compared to other culture free methods, the Raman spectroscopy combined with CNN offers a faster and easier interpretation of the samples, thus reducing the risk of antimicrobial resistance development that comes with empirical treatment [40].

Brincat and Hofmann created a text-mining system that incorporates DL algorithms to help speed up the curation process. This kind of algorithm can replace the manual curation process by screening a large amount of literature to find genes related to AR. The new text-mining algorithm was used to identify ARGs of *Helicobacter pylori* (*H. pylori*), a bacterium that infects the stomach of more than half of the world’s population. The algorithm was then tested by using the nitroimidazole antibiotic group as a case study. Results showed that out of 28 identified genes, 23 should be included in knowledge databases because they could serve as new candidates in studies regarding *H. pylori* resistance to antibiotics [41].

Ciloglu et al., used, for the first time, a new bacterial resistance identification technique to distinguish between MRSA and methicillin-susceptible *Staphylococcus aureus* (MSSA). This new method combined surface-enhanced Raman spectroscopy (SERS) with DL algorithms. Spectral data was acquired by illuminating the bacteria coated by silver nanoparticles (AgNPs) and authors anticipated spectral differences between MRSA and MSSA cell wall. The data was then processed by an autoencoder to train the DL model, represented by a CNN. Authors stated that SERS successfully identified variations between the MRSA and MSSA surface that could indicate the presence of structures related to AR, such as modified penicillin binding proteins (PBPs). In conclusion, SERS technique combined with deep-learning processes could be a valuable tool in the future that will guide clinicians’ decision making by rapid bacterial diagnoses [38].

Wang et al., developed a new DL-derived ensemble method that was able to predict the ARGs to their class. After training the program with ARGs data acquired from the COALA database (collection of all antibiotic resistance gene databases), the algorithm could identify known sequences and align them before predicting their class (e.g., SULFONAMIDE class; TETRACYCLINE class; BETA-LACTAM class). This method will save the time required for classic antimicrobial susceptibility testing (AST) and reduce the failure rate that comes with the empirical treatment of infections [46].

Jang et al., tested how accurate predictions about ARGs occurrence could be by using three different DL models. These methods were compared to determine which one predicts ARGs occurrence linked with certain changes in the environment (e.g., rainfall, tide, salinity etc.). The focus was on determining the abundance of four ARGs found in bacteria that contaminated the water in a recreational beach area in South Korea. Each model was developed based on both meteorological and aquatic variables as input data. All three models’ predictions were compared to the observed emergence of ARGs and thus “loss” values were calculated. One NN technology had an enhanced performance in detecting single ARGs, whereas another model showed superior performance in predicting multi-ARGs and allowed identification of the importance of the input variables [47].

Research on the detection of ARGs, and the discovery of novel antibiotics and antimicrobial peptides that combat these ARGs, is, with the existing technology, not lucrative. High costs and low feasibility of molecule production are the main reasons why the focus is on investing in already stable molecules by improving their properties rather than producing new ones. Thus, many potentially beneficial innovative drugs are overlooked. This lack of advancement leads to limited existing antibiotics on the market. Even with improved formulas and administration in combinations to maximize their effect, we swiftly reach multi-drug resistant bacteria that cannot be therapeutically managed. These bacteria result from natural resistance mechanisms and inappropriate prophylactic measures in clinical settings, such as unsuitable empirical antibiotic treatment [48], administration of antibiotics without subsequent antibiogram correction, and poor hygiene and sterilization measures.

Matrix-assisted laser desorption/ionization time-of-flight mass spectrometry (MALDI-TOF MS) is another technique that can be used for rapid identifications of antibiotic resistance in clinical medicine. The MALDI-TOF mass spectrometer is a popular MS instrument used in many fields of biology. By using this technique, clinicians can rapidly and precisely identify the genus and the species of many Gram-negative and positive bacteria. Microorganism identification by MALDI-TOF MS works by identifying a characteristic spectrum specific to the species and then matching it with a large database [49]. Additionally, differences in biomass after incubation with antibiotics can be used as a rapid test for antibiotic resistance with rapid detection by MALDI-TOF MS [50]. Nevertheless, DL has forged new possibilities for bringing innovative molecules into the market. DL is a computer system, a subtype of ML and AI, that has so far helped with drug repurposing, such as the discovery of propranolol’s dual recommendation as a beta-blocker and treatment of infantile hemangioma [34]. Numerous AMPs were discovered using DL variations, such as the multi-task method [22] and DL modified for mining [41]. DL has also been combined with other methods to increase its accuracy. For example, spectroscopy combined with DL was used to discover new AMPs [40]. In addition, QSAR and DL were associated to differentiate between MRSA-MSSA, prompting a bacterial diagnosis that assists clinical decisions [38].

ML and DL are both types of AI. While ML is AI that can automatically adapt with minimal human expert intervention, DL is a subset of ML that uses ANNs to mimic the learning process of the human brain. We mention that this narrative review presents only the applications of DL in the field of antibiotic resistance.

## 5. Conclusions

Although AR has been used by bacteria for millions of years, in the last decades it has had an increasingly negative impact on our society. Social, economic, and scientific factors contribute to the continuous growth of AR and to its associated problems. Classical approaches to antibiotics development are lengthy and costly, thus new and more efficient techniques must be used in order to select the most promising molecules with innovative mechanisms to which bacteria are naïve. AI can shorten the preclinical phase of drug development by rapidly suggesting many new molecules based on algorithms created by ML techniques such as NN or DL. However, although advancements are continuously made in this domain, the transfer to the clinical context is yet to be achieved. For now, prophylactic measures are the best accessible means of combating AR. By implementing rigorous protocols that stop unnecessary antibiotics prescription, improving hygiene and sterilization procedures, prescribing empirical antibiotic therapy according to the local resistance geographic mapping, and lowering the use of antibiotics in livestock as growth promoters, this prophylaxis can be achieved.

## Figures and Tables

**Figure 1 antibiotics-11-01674-f001:**
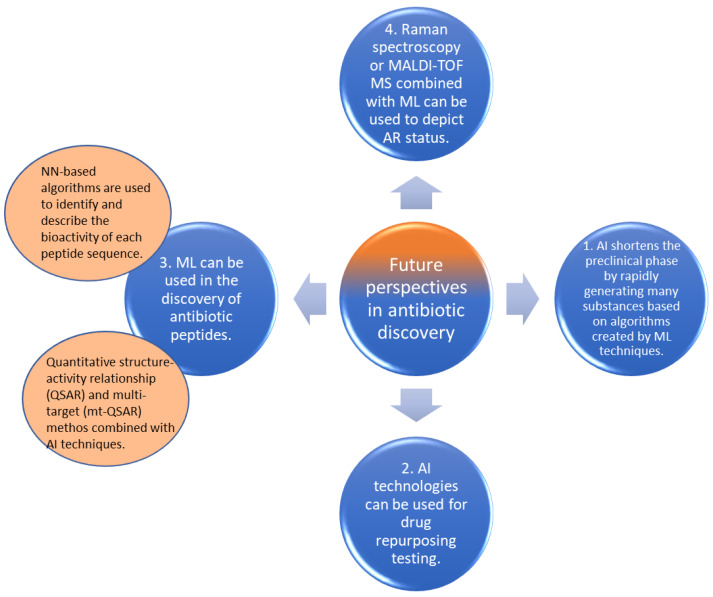
Future perspectives of antibiotic discovery using AI technologies.

## Data Availability

Not applicable.

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
