# Peer review of "Deep Learning and Antibiotic Resistance"

_antibiotics, 2022, doi:10.3390/antibiotics11111674_

Round 1
Reviewer 1 Report
The authors have based their manuscript on a topic of great importance in the context of the application of machine learning approaches in antimicrobial research, with an emphasis on antibiotic resistance. However, the manuscript needs major revisions before being accepted for publication according to the following criteria:
1) Deep learning, despite its wide applications is unlikely to represent the solution to tackle antibiotic resistance from the point of view of computer-aided drug discovery methods. In my opinion, the manuscript should focus on machine learning techniques rather than deep learning.
2) To date, there are several ways in which deep learning and/or machine learning methods can tackle antibiotic resistance:
a) Enabling the virtual design and/or prediction of multi-target inhibitors for proteins present in a given bacterium. Notice that a multi-target inhibitor may be associated with a lower propensity to the appearance of antibiotic resistance.
b) Computer-aided discovery of multi-strain inhibitors. Generally, a molecule able to inhibit many different strains of the same bacterium will have a lower propensity than a molecule that inhibits a few strains.
c) Multi-target inhibitors of different bacteria and strains. This is an extension of the previous point. However, The ADME properties of the designed, discovered, or virtually screened molecules can also be predicted by the same model that predicted antibacterial activity against multiple bacteria and strains.
All these ideas have already been accomplished by different variants of advanced chemoinformatics models that combine the QSAR methodology with diverse machine learning techniques in the context of antibacterial research. These models are known as mt-QSAR (multi-target QSAR), mtc-QSAR (multi-condition QSAR), mtk-QSBER (multi-tasking model for quantitative structure-biological effect relationships), or PTML (perturbation theory and machine learning). All the advances presented here are ways to deal with antibiotic resistance from a computational point of view. Therefore, I advise you to create in your manuscript a new section devoted to all these advanced models and their applications in antibacterial research.
Author Response
The authors thank the reviewer for his/her valuable observations. Here we provide a point-to-point answer to all the author's observations.
Point 1: Deep learning, despite its wide applications is unlikely to represent the solution to tackle antibiotic resistance from the point of view of computer-aided drug discovery methods. In my opinion, the manuscript should focus on machine learning techniques rather than deep learning.
Response 1: Thank you for the suggestion. Machine learning and deep learning are both types of artificial intelligence. While machine learning is AI that can automatically adapt with minimal human interference, deep learning is a subset of machine learning that uses neural networks to mimic the learning process of the human brain. The aim of this narrative review is to present the role of deep learning applications using neural networks in the field of antibiotic resistance. The role of machine learning applications in the field of antibiotic resistance is another topic. Further, we already found in the available medical literature two reviews about machine learning and antibiotic resistance:
Liu G, Stokes JM. A brief guide to machine learning for antibiotic discovery. Curr Opin Microbiol. 2022 Oct;69:102190. doi: 10.1016/j.mib.2022.102190. Epub 2022 Aug 10. PMID: 35963098.
Anahtar MN, Yang JH, Kanjilal S. Applications of Machine Learning to the Problem of Antimicrobial Resistance: an Emerging Model for Translational Research. J Clin Microbiol. 2021 Jun 18;59(7):e0126020. doi: 10.1128/JCM.01260-20. Epub 2021 Jun 18. PMID: 33536291; PMCID: PMC8218744.
Point 2: To date, there are several ways in which deep learning and/or machine learning methods can tackle antibiotic resistance:
- a) Enabling the virtual design and/or prediction of multi-target inhibitors for proteins present in a given bacterium. Notice that a multi-target inhibitor may be associated with a lower propensity to the appearance of antibiotic resistance.
- b) Computer-aided discovery of multi-strain inhibitors. Generally, a molecule able to inhibit many different strains of the same bacterium will have a lower propensity than a molecule that inhibits a few strains.
- c) Multi-target inhibitors of different bacteria and strains. This is an extension of the previous point. However, The ADME properties of the designed, discovered, or virtually screened molecules can also be predicted by the same model that predicted antibacterial activity against multiple bacteria and strains.
All these ideas have already been accomplished by different variants of advanced chemoinformatics models that combine the QSAR methodology with diverse machine learning techniques in the context of antibacterial research. These models are known as mt-QSAR (multi-target QSAR), mtc-QSAR (multi-condition QSAR), mtk-QSBER (multi-tasking model for quantitative structure-biological effect relationships), or PTML (perturbation theory and machine learning). All the advances presented here are ways to deal with antibiotic resistance from a computational point of view. Therefore, I advise you to create in your manuscript a new section devoted to all these advanced models and their applications in antibacterial research.
Response 2: Thank you very much for your observation. We have added more relevant information regarding the suggested technologies:
Generally, a molecule able to inhibit many different strains of the same bacterium will have a lower propensity than a molecule that inhibits a few strains. By following this thread, new QSAR approaches such as the multi-tasking model for quantitative structure-biological effect relationships (mtk-QSAR), also called multi-target (mt-QSAR), are used to integrate different kinds of chemical and biological data, allowing the assessment of multiple biological activities against diverse biological systems. By screening large amounts of data, the descriptors can show which pattern is often associated with the desired conditions (for instance high antimicrobial activity against multiple gram-negative bacteria and low cytotoxicity to human cells). In the end, known peptides are prioritized based on their similarity with these descriptors and other peptides can also be generated with the established set of rules. Only the combination of certain amino acids and the topological distances between them are essential for improving the antibacterial activity [44]. In the study conducted by Kleandrova et al., the study group generated a library formed by 10 peptides, all which exhibited high antibacterial activity against gram negative bacteria based on the molecular descriptors which they used to screen a data set containing 3592 peptides [44].
To this extent, the use of in silico models based on perturbation theory concepts and machine learning technologies (PTML) could measure the probability of a drug being active under certain conditions (protein, cell line, organism) [45]. The combination between PTML and mtc-QSAR models showed promising results in discovering multi-strain inhibitors. In another study, Kleandrova et al. showed that this approach proved to be helpful in identifying molecules that could extend antituberculosis (anti-TB) effect. Twelve molecular descriptors where chosen and their tendencies of variation were calculated, meaning how much these descriptors should vary in order for a molecule to enhance its anti-TB activity and versatility (ability to inhibit more than one Mycobacterium tuberculosis strain). For example, one descriptor analyzes the augmentation of the molecule brought by the hydrophobicity of any two atoms which are located three bonds from one another. The mtc-QSAR-EL model identified proven antituberculosis drugs after screening a dataset made up of 8898 agency regulated chemicals (and investigational FDA-approved drugs). It also recommended compounds with high potential to be experimentally repurposed as antituberculosis (multi-strain inhibitors) agents [45].
Reviewer 2 Report
Deep Learning and Antibiotic Resistance is a very good in demand review work done by the authors. To enhance the work done and make it more readable and of interest to the researchers the following points need to be considered by the authors:
1.At many places syntactic errors are found and sentences are not vey clear in the Abstract & Introduction section. It needs a through revisit of the content.
2. The Objective/Problem definition and contribution is not clearly mentioned in the Abstract
3. The section Present & Future perspective need to be written in more structured form for easy understanding and maintaining the flow. It can be broken in different subsections and some visual representations should be added for better and quick reference.
4. A section highlighting the Critical findings & future Work need to be added
Author Response
The authors thank the reviewer for the careful evaluation of the manuscript. Here we provide a point-to-point response to the reviewer's observations.
Point 1.At many places syntactic errors are found and sentences are not very clear in the Abstract & Introduction section. It needs a through revisit of the content.
Response 1: We have revised the manuscript and have made revisions throughout the manuscript.
Point 2. The Objective/Problem definition and contribution is not clearly mentioned in the Abstract
Response 2: The reviewer is right, we have included a phrase in the Abstract, clearly stating the aim of the review.
The aim of the present review is to highlight the techniques that are being developed for the identification of new antibiotics to assist this lengthy process, using artificial intelligence (AI).
Point 3. The section Present & Future perspective need to be written in more structured form for easy understanding and maintaining the flow. It can be broken in different subsections and some visual representations should be added for better and quick reference. Future perspective
Response 3: We have made modifications to the Present & Future perspective chapters in order to enhance clarity and also for this purpose we have included subsections.
Point 4. A section highlighting the Critical findings & future Work need to be added
Response 4: We have included in the Future perspectives section, subsections highlighting Critical findings and Future work.
Reviewer 3 Report
The authors aimed to provide a narrative review to present the role of deep learning applications in the field of antibiotic resistance. They did integrate a lot of works that related to this issue. However, I have some major concerns to the manuscript.
1) The writing needs to be significantly improved. For example, there are lots of paragraphs which contain only one sentence. Moreover, some sentences are too long to understand the meanings, such as “AI is still used today in this area, and it has been evolving considerably but the new emerging AI is also a powerful ally and has shown promising results in developing new antibiotics in the modern era [11].” (Line 207-209).
2) The authors should pay attention to the consistency of the format. For example, in Line 60, the references are after the dot. While, the in Line 62, the reference is before the dot. The consistency should by double checked.
3) Please provide the full names for all the abbreviations in the first time. Also, the writing formats should be consistent.
4) The writing for the names of bacteria should be consistent. For example, in Line 120, the authors used “E. Coli”, while it was “Escherichia coli” in Line 134.
5) In addition to Raman spectroscopy, there are other techniques using AI such as matrix-assisted laser desorption/ionization time-of-flight mass spectrometry (MALDI-TOF MS) for rapid identifications of antibiotic resistance in clinical medicine. Why the authors did not mention such applications?
6) The authors should provide more indications for the mentioned works. In other words, are there any relationships among these works? It would be more valuable if the authors could provide more explanations.
Author Response
The authors thank the reviewer for the careful evaluation of the manuscript. Here we provide a point-to-point response to the reviewer's observations.
Point 1: The writing needs to be significantly improved. For example, there are lots of paragraphs which contain only one sentence. Moreover, some sentences are too long to understand the meanings, such as “AI is still used today in this area, and it has been evolving considerably but the new emerging AI is also a powerful ally and has shown promising results in developing new antibiotics in the modern era [11].” (Line 207-209).
Response 1: Thank you for your observation. We have modified the manuscript accordingly.
Point 2: The authors should pay attention to the consistency of the format. For example, in Line 60, the references are after the dot. While, the in Line 62, the reference is before the dot. The consistency should by double checked.
Response 2: We have revised the entire manuscript and made consistent changes, evening out the content.
Point 3: Please provide the full names for all the abbreviations in the first time. Also, the writing formats should be consistent.
Response 3: We have revised the manuscript and made the necessary changes.
Point 4: The writing for the names of bacteria should be consistent. For example, in Line 120, the authors used “E. Coli”, while it was “Escherichia coli” in Line 134.
Response 4: Thank you for your observation. We have made the changes accordingly.
Point 5: In addition to Raman spectroscopy, there are other techniques using AI such as matrix-assisted laser desorption/ionization time-of-flight mass spectrometry (MALDI-TOF MS) for rapid identifications of antibiotic resistance in clinical medicine. Why the authors did not mention such applications?
Response 5: Thank you for the observations made. We included in the manuscript the following paragraph.
Matrix-assisted laser desorption/ionization time-of-flight mass spectrometry (MALDI-TOF MS) is another technique that can be used for rapid identifications of antibiotic resistance in clinical medicine. MALDI-TOF mass spectrometer is a popular MS instrument used in many fields of biology. By using this technique, clinicians can rapidly and precisely identify the genus and the species of many Gram-negative and -positive bacteria. Microorganism identification by MALDI-TOF MS works by identifying a characteristic spectrum specific to the species and then matching it with a large database [49]. Additionally, differences in biomass after incubation with antibiotics can be used as a rapid test for antibiotic resistance with rapid detection by MALDI-TOF MS [50].
- Tsung-Yun H., Chuan C.-N, Shih-Hua T. Current status of MALDI-TOF mass spectrometry in clinical microbiology. J Food Drug Anal Jan 2019; 27(2): 404–414.
- Idelevich E. A., Becker K. Matrix-Assisted Laser Desorption Ionization-Time of Flight Mass Spectrometry for Antimicrobial Susceptibility Testing. J Clin Microbiol Nov 2021; 59(12):e0181419.
Point 6: The authors should provide more indications for the mentioned works. In other words, are there any relationships among these works? It would be more valuable if the authors could provide more explanations.
Response 6: The authors thank the reviewer for the observation. Modifications have been included in the manuscript providing more details and explanations to the applications used for AI in antibiotics resistance detection.
Round 2
Reviewer 1 Report
The authors have made the appropriate modification to the manuscript. In my opinion, the manuscript can be accepted for publication.
Author Response
The authors thank the reviewer for the feedback.
Reviewer 2 Report
The revised version has taken into consideration most of the raised concerns, certain minor observations may further improve the readability of the work:
1. In section 4 inspite of using the FUTURE WORK in each sub sections can be avoided by giving a brief para under section4.
2. Certain visual statistics being added in Section 3 and 4 will further improve the readability
Author Response
The authors thank the reviewer for the careful evaluation of our manuscript.
According to the reviewer's observations, we have deleted "Future work" from the subtitles of the "Future perspectives" section and have included an introductory paragraph: "Due to the increasing number of multi-resistant bacteria, new approaches regarding antibiotics development should be considered. Based on some important progresses made in the recent years, beginning with the introduction of AI in antibiotics development, some new areas of antibiotic discovery have started to yield encouraging results."
Moreover, to help with text readability, we have included Figure 1 (Future perspectives of antibiotic discovery using AI technologies).
Reviewer 3 Report
The authors have made a great effort to address the comments and to improve their manuscript. However, the contributions of this work still not clear. The writing formats still have some problems. For instance, the names of the microorganisms should be italicized as the scientific style. Specifically, the genus name is capitalized, and the species is lower case, and hence it should be “Escherichia coli (E. coli)”.
Author Response
The authors thank the reviewer for the observations made. We have verified and corrected the names of the microorganisms, as suggested. Moreover, we have included a small introduction to section 4 and a figure summarising the most important aspects of the section in order to increase the readability of section 4.
Round 3
Reviewer 3 Report
The authors have made a great effort to address the comments and to improve their manuscript. It is appreciated. I have no more concerns.